# Bispecific Antibodies in Multiple Myeloma: Opportunities to Enhance Efficacy and Improve Safety

**DOI:** 10.3390/cancers15061819

**Published:** 2023-03-17

**Authors:** Dawn Swan, Philip Murphy, Siobhan Glavey, John Quinn

**Affiliations:** RCSI Beaumont Cancer Centre, D09V2N0 Dublin, Ireland

**Keywords:** myeloma, bispecific antibodies, T-cell engagers, tumour microenvironment

## Abstract

**Simple Summary:**

Multiple myeloma, a cancer of the bone marrow, is the commonest cancer of adults in the Western World. Therapies have advanced dramatically in recent years, equating to improved survival and quality of life for patients, but those with resistant disease still have less favourable outcomes. Bispecific antibodies represent a new treatment option for patients with myeloma. These antibodies activate the patient’s own T-cells to kill their tumour cells and have shown impressive results in relapsed refractory myeloma. In this review, we consider ways to improve the activity of these new therapies, as well as to reduce the risk of serious side effects. Bispecific antibodies are immune treatments. Given that the immune system is defective in myeloma, we discuss combining bispecific antibodies with other treatments to improve T-cell function in these patients. We also consider how to reduce the risk of cytokine release syndrome, an important side effect of these therapies.

**Abstract:**

Multiple myeloma (MM) is the second most common haematological neoplasm of adults in the Western world. Overall survival has doubled since the advent of proteosome inhibitors (PIs), immunomodulatory agents (IMiDs), and monoclonal antibodies. However, patients with adverse cytogenetics or high-risk disease as determined by the Revised International Staging System (R-ISS) continue to have poorer outcomes, and triple-refractory patients have a median survival of less than 1 year. Bispecific antibodies (BsAbs) commonly bind to a tumour epitope along with CD3 on T-cells, leading to T-cell activation and tumour cell killing. These treatments show great promise in MM patients, with the first agent, teclistamab, receiving regulatory approval in 2022. Their potential utility is hampered by the immunosuppressive tumour microenvironment (TME), a hallmark of MM, which may limit efficacy, and by undesirable adverse events, including cytokine release syndrome (CRS) and infections, some of which may be fatal. In this review, we first consider the means of enhancing the efficacy of BsAbs in MM. These include combining BsAbs with other drugs that ameliorate the effect of the immunosuppressive TME, improving target availability, the use of BsAbs directed against multiple target antigens, and the optimal time in the treatment pathway to employ BsAbs. We then discuss methods to improve safety, focusing on reducing infection rates associated with treatment-induced hypogammaglobulinaemia, and decreasing the frequency and severity of CRS. BsAbs offer a highly-active therapeutic option in MM. Improving the efficacy and safety profiles of these agents may enable more patients to benefit from these novel therapies and improve outcomes for patients with high-risk disease.

## 1. Introduction

Multiple myeloma (MM) is the second most common haematological malignancy of adults in the Western world, with increasing rates reported over recent years [1,2]. MM is characterized by the clonal expansion of neoplastic plasma cells, leading to the production of a paraprotein, anaemia, renal impairment, bone damage, and humoral and cellular immunosuppression [3,4]. Outcomes have improved significantly since the advent of proteosome inhibitors (PIs), immunomodulatory agents (IMiDs), and monoclonal antibodies. The median overall survival (OS) has doubled to approximately 5 years [5]. However, patients with adverse cytogenetics or high-risk disease as determined by the Revised International Staging System (R-ISS) continue to have less durable responses to treatment, and the majority of low-risk patients will eventually develop treatment-resistant disease [6,7,8].

Patients with ultra-high risk disease, so-called ‘double-hit’ myeloma, defined by the presence of biallelic inactivation of TP53 or 1q21 amplification and ISS stage III disease, typically succumb to their disease within 2 years [9]. Those who are refractory to the 3 classes of novel agents (triple-refractory) have an OS of less than one year, whereas the median OS for penta-refractory patients (refractory to 2 PIs, 2 IMiDs and a monoclonal antibody) is a dismal 5.6 months [10]. Given this unmet need, novel therapies remain a priority in MM.

## 2. Overview of Bispecific Antibodies in Myeloma

Bispecific T-cell antibodies (BsAbs) are designed to simultaneously bind to a target moiety on tumour cells and to CD3 on T-cells. This causes direct T-cell activation and subsequent tumour cell killing [11,12]. The earliest BsAbs consisted of fragment antigen-binding (Fab) variable regions connected by a short flexible linker (non IgG-like BsAbs). Such small BsAbs have a short half-life and require continuous intravenous infusion. Newer agents include a fragment crystallizable (Fc) region (IgG-like BsAbs). These larger BsAbs can be administered via intermittent infusion or subcutaneous (S/C) injection [13].

Blinatumumab, the first licensed BsAb, is a CD19-directed non IgG-like construct that was approved for use in acute lymphoblastic leukaemia (ALL) in 2014 [14]. Since then, numerous BsAbs have been developed in a variety of conditions, including MM. Various targets on malignant plasma cells are being investigated, with the majority of work to-date focused on B-cell maturation antigen (BCMA) [15]. BCMA is a member of the tumour necrosis family receptor superfamily, expressed by mature B-cells, plasma cells and MM cells [16,17,18,19]. It has roles in MM cell survival through the upregulation of anti-apoptotic proteins [20,21,22]. Levels of soluble BCMA (sBCMA) increase with disease progression and correlate with adverse outcomes [23,24]. In 2022, two anti-BCMA BsAbs received regulatory approval. Teclistamab was approved by the European Medicines Agency (EMA) and the Food and Drug Administration (FDA) for use in relapsed refractory myeloma (RRMM), and elranatamab also received orphan drug designation by both the EMA and FDA [25,26].

Teclistamab is a humanized IgG Fc anti-BCMA BsAb. Regulatory approval was granted following the publication of results of the phase 1/2 MajesTEC-1 study (NCT03145181) in 165 RRMM patients with triple class-exposed disease. After a median follow-up of 14 months, the overall response rate (ORR) was 63%, and 39% achieved a complete response (CR) or better. The median duration of response (DOR) was 18 months, with a median progression-free survival (PFS) of 11 months. CRS occurred in 72%, immune effector cell-associated neurotoxicity syndrome (ICANS) in 3%, and infections in 76%, of which 45% were grade 3–4 events [27].

Elranatamab is a humanized IgG2A anti-BCMA BsAb. The phase 2 MagnetisMM-3 study (NCT04649359) enrolled and treated 123 RRMM patients. A total of 97% of the trial population were triple-refractory, and 42% were penta-refractory. After a median follow-up of nearly 7 months, the ORR was 61%. Some 51% of patients were still receiving elranatamab at data cut-off, with progressive disease accounting for 33% of those discontinuing therapy. Of 119 patients, CRS occurred in 56% and ICANS in 3%. Infections were reported in 62%, 32% being of grade 3–4 [28]. A summary of the published data for the reported BCMA BsAbs in MM in shown in Table 1.

## 3. Improving Efficacy

One of the hallmarks of MM is the tumour-permissive microenvironment (TME) [15]. A complex interplay between MM cells, immune cells and bone marrow stromal cells (BMSCs) impairs normal immune function, facilitating MM cell proliferation and survival.

Of concern to BsAb efficacy, effector T-cell function is particularly compromised in MM, with progressive dysfunction observed with disease progression and increasing treatment lines [29]. For example, BM-derived T-cells from MM patients fail to mount a robust cytokine response when exposed to MM cells, compared with T-cells from patients with monoclonal gammopathy of uncertain significance (MGUS) [30]. A number of mechanisms underpin this dysfunction. MM cells induce T-cell anergy by upregulating the expression of checkpoint ligands and receptors such as programmed death-ligand 1 (PD-L1), which binds to programmed death receptor-1 (PD-1), leading to T-cell exhaustion, impaired cytokine production and reduced target cell lysis [31]. PD-L1 expression is increased on MM cells compared with both MGUS cells and healthy donor plasma cells [32,33].

Immunosuppressive Regulatory T-cells (Tregs) are enriched in MM peripheral blood samples. MM cells themselves can induce the formation of Tregs in vitro, [34], promoting immune escape, and perhaps explaining the increasing levels of Tregs present with increasing disease burden. In a murine model, the depletion of Tregs in mice with established MM promoted vigorous T-cell and NK cell-mediated responses, halting disease progression [35]. T-cell function is also impaired by myeloid-derived suppressor cells, present at 5 times the normal level in MM patients [36].

BsAbs rely upon a robust CD8+ cytotoxic T-cell response. Continuous antigen stimulation can lead to T-cell exhaustion, with resultant resistance to therapy anticipated [37]. In patients who respond to BsAb therapy, a selective expansion of clonotypic tumour-reactive CD8+ T-cells is produced, which replaces exhausted BM T-cells. This is not seen in non-responding patients [38]. An analysis of the MajesTEC-1 study of teclistamab in RRMM also showed that patients who failed to respond to treatment had lower peripheral CD8+ T-cell levels, increased levels of Tregs, and enhanced expression of markers associated with T-cell exhaustion in blood and BM samples. Higher levels of exhausted CD8+ T-cells and Tregs pre-treatment were associated with inferior PFS in this study [39].

Given the impairment of T-effector activity seen in MM, combining BsAbs with therapies which augment T-cell function may provide a means to improve efficacy.

## 4. Cyclophosphamide

Cyclophopshamide is a member of the oxazaphosphorine family of mustard-alkylating agents, used in the treatment of MM for over half a century, both for its alkylator and immunomodulatory properties. Low dose cyclophosphamide augments responses to the CD38-monoclonal antibody Daratumumab by altering macrophage function [40,41]. It also has a number of actions which may enhance BsAb activity [42,43]. In particular, cyclophosphamide depletes Tregs, promoting cytotoxic T-cell activity [44]. Tregs are thought to be particularly sensitive to cyclophosphamide due to their relatively low levels of intracellular ATP compared with effector T-cells [45]. Reduced ATP leads to impaired production of the reducing agent glutathione, which is required to neutralize the toxic metabolites of cyclophosphamide. In addition, Tregs have inferior DNA repair mechanisms compared to effector T-cells, promoting susceptibility to the DNA cross-linking effects of cyclophosphamide [46].

Low-dose cyclophosphamide can improve effector T-cell responses by reducing Treg-mediated immunosuppression, and by skewing T helper cells from a Th2 profile to a Th1 profile. Th1 cells produce interleukin-2 (IL-2), which stimulates the expansion of memory cytotoxic T-cells [47], required for BsAb activity. Increased levels of IL-17 producing proinflammatory CD4+ helper T-cells have also been observed post-cyclophosphamide exposure. The combination of elevated Th17 cells and suppressed Tregs has been associated with improved survival in MM [48].

In a murine model, the addition of cyclophosphamide to BCMA-BsAb therapy led to a reduction in Tregs and increased cytotoxic CD8+ T-cells compared with BsAb alone. Tumour rechallenge caused relapse only in mice who did not receive cyclophosphamide. Furthermore, although the addition of pomalidomide to the BsAb in this model also led to improved T-cell cytolytic activity, the persistence of functional T-cells was only observed in cyclophosphamide-treated animals [49].

At the time of writing, there are no clinical trials combining cyclophosphamide with BsAb therapy in MM, although this represents a promising approach to improving efficacy.

## 5. IMiDs

IMiDs act via binding to cereblon, a component of the E3 ubiquitin ligase. IMiDs in current use include lenalidomide, pomalidomide and thalidomide. Cereblon-binding results in ubiquitination and proteasome-mediated degradation of the Ikaros family transcription factors 1 and 3, with downstream reduced transcription of MYC and IRF4, which are required for survival and proliferation [50]. In addition to their direct effect on MM cells, IMiDs have numerous immunomodulatory capabilities. Exposure to lenalidomide enhances myeloma-specific T-cell responses in vivo and in vitro [51]. Lenalidomide maintenance following allogeneic stem cell transplant in MM increases the incidence and severity of graft versus host disease, suggesting potentiated T-cell responses [52]. Pre-clinical work using the anti-BCMA BsAb AMG-701 (pavurutamab), an extended half-life single chain variable fragment BsAb [53], showed that pre-treatment of immune effector cells with lenalidomide or pomalidomide enhanced AMG-701-mediated MM cell killing [54]. Furthermore, the activity of an anti-BCMA BsAb in a murine MM model was improved by the addition of pomalidomide, leading to increased levels of circulating lytic T-cells, even in lenalidomide-resistant cases, although T-cell exhaustion was noted [49]. Iberdomide and mezigdomide are novel cereblon E3 ligase modulators (CELMoDs) which have more potent tumoricidal and immunogenic effects than the IMiDs and produce greater MM cell killing [55,56]. The asymmetric 2-arm, humanized IgG BsAb alnuctamab [57] was assessed in a preclinical model, combined with pomalidomide, iberdomide or mezigdomide. BsAb-induced MM cell killing was enhanced by all combinations compared with alnuctamab alone, with the greatest effect seen using mezigdomide, particularly when artificially exhausted donor T-cells were used. The combination of mezigdomide and alnuctamab was also shown to enhance T-cell activation, T-cell infiltration of tumour tissue, and tumour clearance in a murine MM model compared with the BsAb alone [58].

These data support combining BsAbs with IMiDs and CELMoDs. Initial results from the phase 1b MajesTEC-2 study of teclistamab with daratumumab and lenalidomide in RRMM have shown promising results with an ORR of 90% and an acceptable safety profile [59]. There are currently several ongoing studies investigating various combinations, as shown in Table 2.

## 6. Checkpoint Inhibitors (CPI)

PD-L1 is upregulated on the MM cell surface, binds to PD-1, and induces T-cell exhaustion [31]. Increased levels of soluble PD-L1 have been shown to be independently associated with shorter PFS in patients following upfront therapy [60]. In B-cell ALL, the potent continuous T-cell activation caused by anti-CD19 BsAbs induces expression of PD-L1 and other immune checkpoint ligands on B-ALL cells [61,62]. In MM, interaction between PD-L1 on MM cells and PD-1 in vitro was shown to impair the efficacy of an anti-Fc receptor homolog 5 (FcRH5) BsAb, whereas PD-L1 blockade improved MM cell killing [63]. High levels of PD-L1 expression by plasma cells were associated with poor responses to the anti-BCMA BsAb AMG-420 in a first-in-human study of RRMM [64]. High levels of PD-1 expressing T-cells are also associated with inferior MM cell lysis following in vitro treatment with the BCMA CAR-T cell Cilta-cel [65].

Prior to the advent of BsAbs in MM, a phase 2 study of the PD-1 antagonist pembrolizumab combined with pomalidomide and dexamethasone yielded promising results in RRMM, prompting further investigation of CPI combinations in two phase 3 studies [66]. KEYNOTE-185 randomised newly diagnosed MM patients to pembrolizumab/lenalidomide/dexamethasone or lenalidomide/dexamethasone, whereas KEYNOTE-183 randomised RRMM patients to pembrolizumab/pomalidomide/dexamethasone or pomalidomide/dexamethasone. Disappointingly, KEYNOTE-185 was halted early due to increased mortality in the investigational arm [67], and ORR was inferior in the investigational arm of KEYNOTE-183 [68]. Pembrolizumab is currently being studied in patients following relapse after BCMA CAR-T cell therapy (NCT05191472) [69]. This study follows on from the observation that pembrolizumab promoted the re-expansion of CAR-T cells in patients with B-cell lymphomas who had failed CD19 CAR-Ts [70]. Another group has generated a novel trispecific antibody (CDR101) which recognizes BCMA, CD3 and PD-L1. This antibody induced stronger and more durable responses than a BCMA BsAb control in a murine model of MM [71]. In the clinical setting, the TRIMM-3 study is a non-randomised phase 1b study in RRMM, where patients receive either teclistamab or the G protein-coupled receptor, class C, group 5, member D (GPRC5D)-directed BsAb talquetamab, in combination with a PD-1 inhibitor (NCT05338775). This study is open to recruitment.

## 7. Earlier Use within the Treatment Paradigm

Immune dysfunction in MM is progressive, impacted by both disease features and increasing lines of therapy [72,73]. Earlier use of T-cell directed therapies may therefore lead to improved responses. The expansion and activity of a BCMA CAR-T was improved in the presence of an early memory T-cell phenotype and a preserved CD4:CD8 T-cell ratio, both of which are more commonly seen in leukapheresis products from patients post-frontline therapy rather than in RRMM [74]. The T-cell environment immediately following ASCT may also be uniquely suited to BsAbs and CAR-T cell therapy, in that Tregs are suppressed while cytotoxic CD8+ T-cells recover, leading to a markedly low Treg:CD8 T-cell ratio that may favour BsAb activity [75]. Published reports regarding BsAbs currently only pertain to use in the RRMM space. Results from several ongoing trials in NDMM, all using BCMA-directed BsAbs, are anticipated, as shown in Table 3.

## 8. Improving Antigen Availability

Resistance to BCMA-directed therapies is incompletely understood. In addition to progressive immune-paresis in MM, the reduction or loss of BCMA expression may also lead to resistance. Gamma-secretase cleaves BCMA from the MM cell surface, producing sBCMA, and potentially reducing target-availability for BsAb binding. Analysis from the MajesTEC-1 study of teclistamab found that the baseline surface expression of BCMA was highly variable and did not correlate with response, but that increased sBCMA levels were associated with poorer responses to treatment [39]. The addition of a gamma secretase inhibitor in vitro enhances BCMA BsAb-induced MM cell killing, with rapid clearance of circulating sBCMA, enhanced tumour control, and improved survival demonstrated in a murine model [76]. Similar results have been shown with BCMA CAR-T cell therapy, although clinical data is not yet available [77].

Rare events of biallelic deletions or mutations affecting the BCMA gene (TNFRSF17) have been predominantly reported following CAR-T cell therapy, and occasionally in the context of BsAbs [78,79,80]. Whole genome sequencing of 100 MM patients identified that 4% of patients had a pre-existing heterozygous aberration in TNFRSF17. Certain patients may therefore be susceptible to BCMA target loss when treated with BCMA-directed therapies [80]. Approaches to the management of patients with antigen-escape include the use of T-cell directing therapies with different targets, combining therapies specific to multiple epitopes or using more than one BsAb simultaneously.

Other antigens under development as potential targets in MM include GPRC5D, a transmembrane orphan receptor of the G protein-coupled receptor family [81], FcRH5, a membrane protein, which regulates B-cell receptor signaling [82], and CD38, a member of the ADP-ribosyl cyclase family, which is involved in the regulation of calcium homeostasis, signaling and adhesion [83]. Talquetamab is an IgG4 Fc BsAb that is specific for GPRC5D. A phase 1 dose-finding study (MonumenTAL-1) of RRMM patients who had progressed with established therapies reported an ORR of 64–70% for 74 patients receiving either 405 μg/kg S/C weekly or 800 μg/kg S/C alternate-weekly, including patients who had received prior BCMA therapy. CRS occurred in 77–80% of patients and was grade 1–2 in the vast majority [84]. Talquetamab is being assessed in a variety of combinations in the RRMM setting (MonumenTAL-2, NCT05050097; MonumenTAL-3, NCT05455320; MonumenTAL-5, NCT05461209), and as a dual antigen-targeting approach combined with teclistamab in the RedirecTT-1 study (NCT04586426). GPRC5D is also under investigation as a target for CAR-T cell therapy in RRMM [85,86].

Cevostamab is a humanized IgG1 Fc BsAb directed against FcRH5. Results from 16 RRMM patients for whom no established treatment options remained, demonstrated a best response of CR or greater in 10 patients, with eight patients maintaining a response 6 months or more following completion of the planned 17 cycles of therapy NCT03275103 [87]. The results from the phase 1b CAMMA-3 study of cevostamab monotherapy, cevostamab with pomalidomide/dexamethasone, or cevostamab with daratumumab/dexamethasone in RRMM are awaited (NCT04910568). Trispecific antibodies are also under development. ISB 2001 simultaneously targets 2 MM epitopes, namely BCMA and CD38, alongside activating T-cells via CD3-binding. Promising results were obtained in a murine MM model [88]. Another trispecific antibody, ISB 1442, targets CD38, CD47 and CD3. In vitro work has shown improved MM cell killing through both T-cell activation and enhanced macrophage-mediated cellular phagocytosis via binding to CD47 [89].

Neoantigens are novel protein sequences produced by cancer cells due to the acquisition of somatic mutations, which can promote an immune response. Increased expression of neoantigens in MM patients following relapse has been demonstrated. One group showed that use of immunological therapies was associated with increased detection of neoantigen-specific T-cells, which correlated with response to therapy [90]. Personalised neoantigen-vaccines can be created by profiling patients at relapse. Such vaccines are under investigation in RRMM and solid tumours [91]. Whether BsAbs could be generated to successfully target neoantigens has not been studied to date.

A summary of the discussed BsAbs in MM is shown in Figure 1, and approaches to improving BsAb efficacy in MM are shown in Figure 2.

## 9. Safety

### 9.1. Infection

As we accumulate data regarding BsAbs in MM, there is growing evidence that these therapies are associated with high rates and severity of infections. Results from the two approved BsAbs in heavily pre-treated immunocompromised patients suggest an overall infection rate of 62–76%, with 32–45% grade 3–4 events seen [27,28]. A meta-analysis of 790 patients treated with a BsAb (directed against BCMA in 73%) in 10 trials reported infections in 44%, of which 26% were grade 3–4, with a short follow-up of less than 5 months. A total of 15% of patients developed COVID-19 infection (11% grade 3–4), and opportunistic infections were also seen, including cytomegalovirus infection (CMV) in 4%. Some 49% of the cohort had documented hypogammaglobulinaemia [92]. Another group reviewed 37 patients treated with an anti-BCMA BsAb. This heavily pre-treated cohort received a median of 13 months treatment, with a PFS of 19 months achieved. The ORR was 70%, and all patients who responded developed severe hypogammaglobulinaemia (IgG <2 g/L). All patients received varicella zoster virus (VZV) prophylaxis, and 92% of hypogammaglobulinaemic patients received IV immunoglobulins (IVIg). Antifungals and antibiotics were not used. A total of 118 infections (22% grade 3–5) were reported during 424 months follow-up. Some 46% of infections were viral, 43% were bacterial, and 11% were fungal. Respiratory tract infections accounted for 58% of events, followed by skin and urinary tract infections at 15% each. CMV reactivation was identified in 22%, with two cases of oesophagitis reported, and 43% of patients contracted COVID-19, with one death. Importantly, the use of IVIg reduced the risk of grade 3–5 infections by 80%. The median time to first grade 3–5 infection was approximately 4 months; however the risk of severe infection continued to rise with time on treatment, with no plateau seen in this study [93]. A retrospective study of 62 patients treated with BCMA-directed therapy included 36 BsAb-patients and 26 CAR-T patients. After 9 months follow-up, 41% of the BsAb group had experienced at least one infective episode compared with 23% of the CAR-T group, with a higher infection density also seen in the BsAb group. IVIg was given to approximately a quarter of each cohort; however, the patients receiving BsAbs were more heavily pre-treated, which may have partially contributed to the increased rates of infection seen. Another possible cause for the observed disparity is the continuous nature of BsAb therapy compared with one-off CAR-T cell treatment [94].

The use of BsAbs earlier within the treatment paradigm may feasibly lead to fewer infectious complications in patients with less marked disease- and treatment-related immunosuppression. Early implementation of prophylactic IVIg should be strongly considered alongside VZV prophylaxis. The risk of CMV reactivation appears to be significant, and monthly CMV monitoring should be considered. Pneumocystis jiroveci pneumonia (PJP) has rarely been seen [92], with no clear requirement for PJP prophylaxis currently; however, longer term follow-up of clinical trials and real-world data is eagerly awaited.

### 9.2. CRS

Cytokine release syndrome, or CRS, occurs as a consequence of robust cytotoxic T-cell activation and the subsequent release of inflammatory cytokines, particularly IL-6. A comparison of the rates and severity of CRS from early publications of CAR-T and BsAb therapy in MM reported rates of 21–79% (median 66%), of which very few events were grade ≥3 for BsAbs, compared with a far higher incidence of severe CRS (grade ≥3) following CAR-T cell therapy (4–41%) [15]. The seven studies shown in Table 1 include published data on 681 participants. CRS was reported in 59%, of which 1.3% was grade 3 [27,28,53,64,95,96,97].

There are a number of mechanisms under investigation to reduce the risk of CRS, such as the use of step-up dosing regimens and corticosteroid-premedication. In the MagnetisMM-1 study of elranatamab, no step-up dosing was used during the dose-escalation phase (80–1000 μg/kg), with an overall rate of CRS of 73%, including prolonged events lasting over 10 days in two patients. Within MagnetisMM-3 and -9, different step-up regimens were incorporated with a single priming dose of 44 mg with or without premedications (acetaminophen 650 mg or paracetamol 500 mg, diphenhydramine 25 mg or equivalent and dexamethasone 20 mg or equivalent), or a two-dose step-up regimen with premedications of either 12 mg and 32 mg or 4 mg and 20 mg. Rates of CRS were 100% (all grade 1–2), 78% (45 grade 3), 56% (all grade 1–2) and 60% (all grade 1–2), respectively. The CRS duration was shorter with the two-dose step-up regimens (median two days versus three days for one dose step-up), but CRS events were more likely to occur with later doses of elranatamab using the 4 mg and 20 mg combination [98].

Altering the design of the BsAb such that it binds to CD3 with low affinity leads to tumour-cell killing but reduced cytokine release [99]. In the phase 1 study of the BCMA BsAb ABBV-383, no step-up doses or premedications were administered. CRS occurred in 72% of patients receiving 60 mg of ABBV-383 (2% grade 3–4) [96]. Another means of reducing CRS may be to incorporate additional therapies with immunomodulatory capabilities. Pre-treatment of peripheral blood immune cells with the CELMoD mezigdomide reduced release of IL-6 by 50% following the treatment of BCMA-positive cells with alnuctamab, without impacting secretion of T-cell-derived cytokines. These data suggest that CELMoDs such as mezigdomide could have the potential to ameliorate CRS in patients receiving BsAbs [100].

## 10. Conclusions

BsAbs and other T-cell directing therapies are highly efficacious, even in heavily-pretreated patients. These therapies provide hope for high-risk patients for whom available treatment modalities fail. At the present time, we do not know how to best sequence BsAbs within the existing MM treatment paradigm. Given their reliance upon T-cell mediated immunity, harnessing the TME may prove key to optimizing their utility, as described above. Clinical trials using BsAbs earlier in the disease course, and in combination with additional immunomodulatory agents, are eagerly awaited. The use of technologies such as CRISPR/Cas9 may provide further potential targets for synergism with BsAbs, by identifying genes which regulate CD8+ T-cell responses, such as the mammalian target of rapamycin (mTOR) pathway [101]. Whether specific high-risk groups may derive particular benefit from these treatments, such as those with adverse cytogenetics or extramedullary disease, is another important question which hopefully will be answered by future studies.

## Figures and Tables

**Figure 1 cancers-15-01819-f001:**
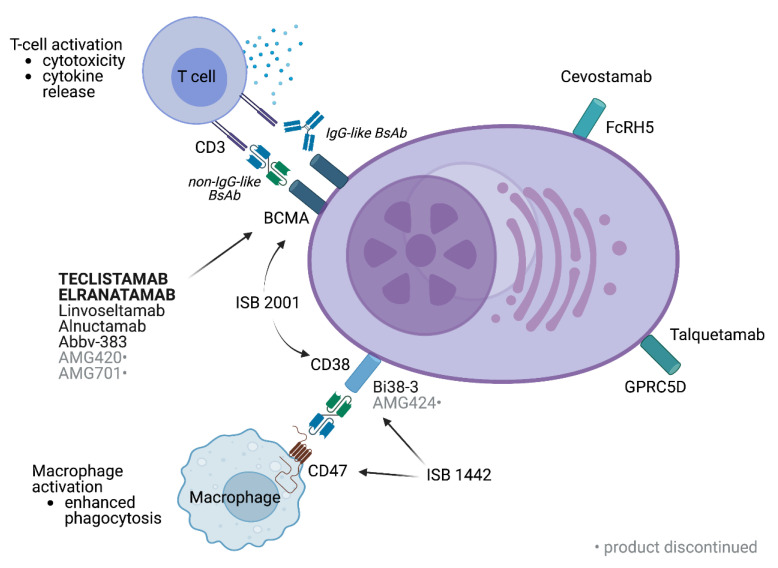
BsAbs under investigation in MM. BsAbs may be IgG-like or non-IgG-like, as shown. The majority of agents target BCMA. Trispecific antibodies such as ISB 2001 may target multiple MM epitopes or enhance immune activation via binding to T cells and macrophages, such as ISB 1442.

**Figure 2 cancers-15-01819-f002:**
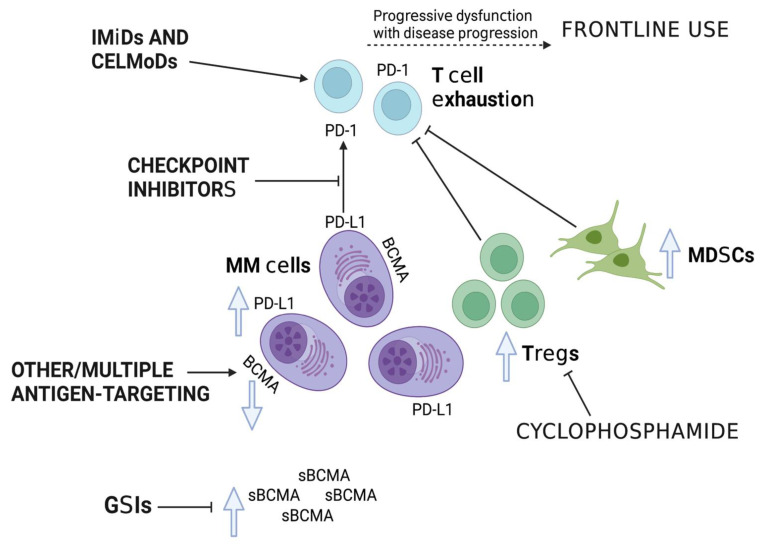
Approaches to harnessing the TME to improve the efficacy of BsAbs in MM.

**Table 1 cancers-15-01819-t001:** Published clinical trials of BCMA BsAbs in RRMM.

Bispecific Antibody	Clinical Trials Identifier	Antibody Structure	Administration	Safety	CRS/ICANS	Responses	Ongoing Studies
Teclistamab	MajesTEC-1 NCT03145181	humanized, IgG Fc	Teclistamab 1.5 mg/kg weekly S/C with a 2-step-up priming dose regimen (0.06 mg/kg and 0.3 mg/kg)	Anaemia 52%, neutropenia 71%, thrombocytopenia 40%, infections 76% (grade 3–4 45%), neurotoxicity 15%	CRS 72% (all but one case grade 1–2), ICANS 3% (all grade 1–2)	ORR 63%, 39% CR or better, median DOR 18.4 months	Several MagesTEC studies ongoing using teclistamab in RRMM and NDMM in combination therapies
Elranatamab	MagnetisMM-3 NCT04649359 Cohort A	full length, humanized, IgG2a	Elranatamab 76 mg weekly S/C on a 28 day cycles with a 2-step-up priming dose regimen (12 mg and 32 mg)	Anaemia 56%, neutropenia 53%, thrombocytopenia 27%, infection 62% (grade 3–4 32%), %, peripheral neuropathy 17%, nausea 30%, diarrhoea 45%	CRS 56% (all grade 1–2), ICANS 3% (all grade 1–2)	ORR 61%, median DOR not reached	Several MagnetisMM studies ongoing using elranatamab in RRMM and NDMM in combination therapies
AMG 420	NCT02514239	BiTE	Continuous 28 day IV infusion followed by 2 week break. Dose-escalation from 0.2–800 μg/day	Infection 33%, polyneuropathy 5%, 12% deranged liver enzymes	CRS 38% (94% Grade 1–2)	ORR 31% across all doses, 70% for the 400 ug/day cohort	Development discontinued by Amgen
AMG 701	NCT03287908	extended half-life, scFvs plus Fc region	Weekly IV. Dose-escalation from 5 μg–12 mg	Anaemia 43%, neutropenia 23%, thrombocytopenia 20%, diarrhoea 31%, fatigue 25%, infection 17%, elevated pancreatic enzymes 3%.	CRS 61% (90% Grade 1–2)	ORR 36% for 3–12 mg doses	Development discontinued by Amgen
Linvoseltamab (REGN5458)	NCT03761108	Fc Fab arms	IV weekly, then every 2 weeks. Dose escalation over 9 dose levels.	Anaemia 37%, neutropenia 29%, thrombocytopenia 21%, fatigue 34%	CRS 48% (all but one case Grade 1–2)	ORR 41% for doses <200 mg and 75% ≥200 mg, median DOR not reached	Phase 2 study of 200 mg REGN5458 is recruiting
Alnuctamab (CC-93269)	NCT03486067	2 arm humanized IgG1 Fc	Dose escalation of IV alnuctamab from 0.15–10 mg. S/C alnuctamab given on D1, 4, 8, 15 and 22 of C1, weekly in C2–3, every other week in C4–6 and every 28 days thereafter. Dose escalation from 10–60 mg	Anaemia 34%, neutropenia 34%	CRS 53% (all grade 1–2), 1 grade 1 ICANS	IV alnuctamab ORR 39%, median PFS 13 weeks, median DOR in responding patients 146 weeks. S/C alnuctamab ORR 51% across all doses, 77% for doses ≥30 mg	Ongoing recruitment to the phase 1 study
Abbv-383	NCT03933735	IgG4 Fc. 2 heavy chain only anti-BCMA moieties	Dose escalation and expansion cohorts (n = 6 in 40 mg cohort, n = 60 in 60 mg cohort)	Infections in 50% of 40 mg cohort and 43% of 60 mg cohort, neutropenia in 67%/40%, anaemia in 33%/32%, thrombocytopenia 33%/25%	CRS 83% (all grade 1–2) in 40 mg cohort and 72% (2% grade 3–4) in 60 mg cohort	ORR 57% across all groups, 83% at 40 mg and 60% at 60 mg. ≥CR 67% at 40 mg and 29% at 60 mg	Phase 1b study planned NCT05650632

**Table 2 cancers-15-01819-t002:** Ongoing studies of a BsAb in combination with an IMiD in RRMM.

Study	BsAb	Target	IMiD-Based Combination Therapy
MajesTEC-2 NCT04722146	Teclistamab	BCMA	Tec/dara/pom, tec/dara/bort/len, tec/dara/len
MajesTEC-7 NCT05552222	Teclistamab	BCMA	Tec/dara/len vs. Dara/len/dex
MagnetisMM-4 NCT05090566	Elranatamab	BCMA	Elran/len/dex
LINKER-MM2 NCT05137054	Linvoseltamab	BCMA	Linvo/len, linvo/pom
NCT04910568	Cevostamab	FcRH5	Cevo/Pom/dex
NCT05050097	Talquetamab	GPRC5D	TalqLen, talqdara/len, talq/pom

**Table 3 cancers-15-01819-t003:** Ongoing studies of BCMA-BsAbs in NDMM.

Study	Design	Treatment	Eligibility
MajesTEC-2 NCT04722146	Multi-arm phase 1b study	Teclistamab with other MM therapies (daratumumab, pomalidomide, lenalidomide, bortezomib, nirogacestat, in various combinations, arms A-F)	Elligibility differs according to treatment arm. Arm B) tec/dara/len/bort (Q21): NDMM or RRMM naïve to lenalidomide. Arm E) tec/dara/len: NDMM or RRMM with 1–3 prior lines including PI/ImID. Arm F) tec/dara/len/bor (Q28): NDMM only.
MajesTEC-4 NCT05243797	Randomised, open-label, multicentre phase 3 study	Tec/len vs. lenalidomide maintenance post ASCT	NDMM patients who have undergone induction and ASCT
MajesTEC-7 NCT05552222	Phase 3 randomised study	Tec/dara/len vs. Dara/len/dex	NDMM patients either ineligible or not suitable for ASCT
MagnetisMM-6 NCT05623020	Open-label, 2 arm, multicentre, randomised study	Elran/dara/len vs. Dara/len/dex	NDMM ineligible for ASCT
MagnetisMM-7 NCT05317416	Randomised, 2-arm, phase 3 study	Elranatamab vs. lenalidomide monotherapy	NDMM patients who are MRD positive post ASCT

## Data Availability

Data sharing not applicable. No new data were created or analysed in this study. Data sharing is not applicable to this article.

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
