# Peer review of "Bispecific Antibodies in Multiple Myeloma: Opportunities to Enhance Efficacy and Improve Safety"

_cancers, 2023, doi:10.3390/cancers15061819_

Round 1

Reviewer 1 Report

The article entitled ' Bispecific antibodies in Multiple Myeloma: Opportunities to enhance efficacy and improve safety’ by Swan et al, greatly describes the current  therapeutic treatments for multiple myeloma. The article is well written and  interesting it sheds light on the latest clinical studies.  However, the authors could improve the article.

1.The introduction of the article is interesting, the authors can include a figure illustrating the binding capabilities of bispecific antibodies and the nexus of T cells and tumor cells.

2. A schematic illustration of the various types of antibodies would be good like the humanized antibodies etc.

3. RRMM and PFS abbreviation needs to be included.

4. The authors can shed light on the latest technologies such as CRISPR and mRNA therapy and neo antigens for the treatment of multiple myeloma.

5.  The authors can include below additional articles.

GPRC5D is a target for the immunotherapy of multiple myeloma with rationally designed CAR T cells

ERIC L. SMITH, et al , 2019

Phase I study of the anti-FcRH5 antibody-drug conjugate DFRF4539A in relapsed or refractory multiple myeloma

Stewart et al, 2019

Author Response

Dear reviewer 1, 

thank you for considering our manuscript. Our responses to your comments are outlined below.

1.The introduction of the article is interesting, the authors can include a figure illustrating the binding capabilities of bispecific antibodies and the nexus of T cells and tumor cells. 

Thank you for this suggestion. We have added an additional figure (figure 1), which summarises the properties of the BsAbs discussed within the paper. We have included the trispecific T-cell and macrophage-activating antibody ISB 1442 in the figure, and briefly into the manuscript within the ‘Improving antigen availability’ paragraph.

  1. A schematic illustration of the various types of antibodies would be good like the humanized antibodies etc.

Thank you for this suggestion. We have included the 2 main types of BsAbs within the newly created figure 1 ie. IgG-like and non-IgG-like. We have not discussed manufacturing elements of BsAbs in detail within the paper or within the figure as this falls outside our particular scope.  

  1. RRMM and PFS abbreviation needs to be included.

RRMM is first used in paragraph 2 of Overview of bispecific antibodies in myeloma where it is explained in the following sentence: ‘Teclistamab was approved by the European Medicines Agency (EMA) and Food and Drug Administration (FDA) for use in relapsed refractory myeloma (RRMM), and Elranatamab also received orphan drug designation by both the EMA and FDA.’

PFS is first used in paragraph 3 of Overview of bispecific antibodies in myeloma where it is explained in the following sentence: ‘Median duration of response (DOR) was 18 months, with a median progression-free survival (PFS) of 11 months.’

  1. The authors can shed light on the latest technologies such as CRISPR and mRNA therapy and neo antigens for the treatment of multiple myeloma. 

Thank you for this suggestion. These new technologies and approaches are certainly interesting, although are in their infancy where bispecific antibody therapies are concerned. We have included a paragraph regarding neoantigens in the ‘improving antigen availability’ paragraph. CRISPR and mRNA therapies are more applicable to CAR-T therapies at the current time. Use of mRNA to produce time-limited BCMA-expression in a CAR-T model may be another mechanism of reducing CRS. This approach is being studied using the Descartes-11 construct. CRISPr/Cas9 can be employed to create allogeneic CAR-Ts. In the context of BsAbs, a CRISPR approach has been used to identify potential regulators of BsAb efficacy in MM, identifying possible targets for synergism. We have mentioned this work within the conclusion.

  1. The authors can include below additional articles.

GPRC5D is a target for the immunotherapy of multiple myeloma with rationally designed CAR T cells. ERIC L. SMITH, et al , 2019

Phase I study of the anti-FcRH5 antibody-drug conjugate DFRF4539A in relapsed or refractory multiple myeloma. Stewart et al, 2019

Thank you for suggesting inclusion of these papers. We have included the citation by Smith et al. along with Mailankody et al. 2022’s paper describing use of an anti-GPRC5D CAR-T cell construct in a phase 1 study. We have not included the paper by Stewart et al. We agree with the reviewer that this is an interesting paper with merit, however the focus of the manuscript is T-cell engaging bispecific antibodies, whereas this study concerns an antibody-conjugate. These treatments have a different set of considerations compared with the BsAbs, and are discussed elsewhere within the literature.

Reviewer 2 Report

General Comment

This is an extensive and timely review on the efficacy and safety of bispecific antibodies in multiple myeloma. The manuscript is well written and balanced.

Specific minor comments

1. I certainly belive that the incidence of myeloma has not increased by 126 percent from 1990 to 2016. Perhaps the data on reference 11 are not accurate. Please double chdck.

2. Please remove Janssen and Pfizer after teclistamab and elranatamab.

3. Number of references are incomplete, please carefully check, particularly the presentation at ASH.

4. Please abbreviate New Englan Journal of Medicine as N Engl J Med.

Author Response

Dear reviewer 2,

thank you for considering our manuscript. We have responded to the comments below. 

  1. I certainly believe that the incidence of myeloma has not increased by 126 percent from 1990 to 2016. Perhaps the data on reference 11 are not accurate. Please double check.

Thank you for this comment. We agree that this statistic does seem surprisingly high, despite being reported in the paper cited. We have removed this from the manuscript to prevent potential confusion.

  1. Please remove Janssen and Pfizer after teclistamab and elranatamab.

We have amended the manuscript as per the suggestion.

  1. Number of references are incomplete, please carefully check, particularly the presentation at ASH.

Thank you for this observation. We have reviewed the references, in particular the ASH abstracts (references 40, 54 and 66) and updated them as appropriate.

  1. Please abbreviate New England Journal of Medicine as N Engl J Med.

Thank you for this comment. We have amended the abbreviation.